# Does Hyperglycemia Cause Oxidative Stress in the Diabetic Rat Retina?

**DOI:** 10.3390/cells10040794

**Published:** 2021-04-02

**Authors:** Mohammad Shamsul Ola

**Affiliations:** Department of Biochemistry, College of Science, King Saud University, 2B10, Building 5, P.O. Box 2455, Riyadh 11451, Saudi Arabia; mola@ksu.edu.sa; Tel.: +966-558-013-579; Fax: +966-146-757-91

**Keywords:** retina, diabetes, hyperglycemia, oxidative stress, metabolism

## Abstract

Diabetes, being a metabolic disease dysregulates a large number of metabolites and factors. However, among those altered metabolites, hyperglycemia is considered as the major factor to cause an increase in oxidative stress that initiates the pathophysiology of retinal damage leading to diabetic retinopathy. Diabetes-induced oxidative stress in the diabetic retina and its damaging effects are well known, but still, the exact source and the mechanism of hyperglycemia-induced reactive oxygen species (ROS) generation especially through mitochondria remains uncertain. In this study, we analyzed precisely the generation of ROS and the antioxidant capacity of enzymes in a real-time situation under ex vivo and in vivo conditions in the control and streptozotocin-induced diabetic rat retinas. We also measured the rate of flux through the citric acid cycle by determining the oxidation of glucose to CO_2_ and glutamate, under ex vivo conditions in the control and diabetic retinas. Measurements of H_2_O_2_ clearance from the ex vivo control and diabetic retinas indicated that activities of mitochondrial antioxidant enzymes are intact in the diabetic retina. Short-term hyperglycemia seems to influence a decrease in ROS generation in the diabetic retina compared to controls, which is also correlated with a decreased oxidation rate of glucose in the diabetic retina. However, an increase in the formation of ROS was observed in the diabetic retinas compared to controls under in vivo conditions. Thus, our results suggest of diabetes/hyperglycemia-induced non-mitochondrial sources may serve as major sources of ROS generation in the diabetic retina as opposed to widely believed hyperglycemia-induced mitochondrial sources of excess ROS. Therefore, hyperglycemia per se may not cause an increase in oxidative stress, especially through mitochondria to damage the retina as in the case of diabetic retinopathy.

## 1. Introduction

Diabetes is an endocrinological disorder that dysregulates several metabolic processes and so forth alters the levels of a multitude of metabolites and signaling molecules, either due to lack of insulin or insulin signaling. Apart from the altered metabolites of carbohydrate, lipids, and amino acids, an increasing number of other biomolecules and hormones including hydroxy acids, pyrimidines, arginine, proline, various peptides, and growth factors have all been found to be altered, making the pathophysiology of diabetes extremely complex [1,2,3,4,5,6]. However, over the years, an increasing amount of research has been dedicated to diabetes-induced hyperglycemia, the hallmark of diabetes as the major factor involved in the etiology of diabetes and its complications [7,8,9,10,11]. Hyperglycemia has been widely considered as the main trigger that initiates the dysregulation of various anabolic and catabolic pathways within cells, thereby inducing cellular damage that leads to various complications of diabetes including diabetic retinopathy, the leading cause of blinding disease worldwide [4,5,12].

Numerous studies have reported mechanisms of hyperglycemia stimulated biochemical abnormalities in the diabetic retina including stimulation of protein kinase C, glycation, polyol formation, and hexosamine synthesis that induce oxidative stress, ultimately leading to cellular damage [10,13,14,15]. Besides, many investigators reported that diabetes-induced hyperglycemia stimulates glycolysis and tricarboxylic acid cycle fluxes that increase NADH/NAD^+^ ratios both in the cytosol and mitochondria of the cells [8,9,16,17,18,19]. This in turn increases electron disposal at the electron transport chain which, thereby produces superoxide radicals by partial reduction of oxygen [20,21]. These findings were partly supported by an increased level of reactive oxygen species (ROS) found in the retina of diabetic animals [8,22], and also in isolated Müller and endothelial cells once exposed to hyperglycemic conditions [23]. In contrast, we and others did not support the hyperglycemic induced fluxes that may generate surplus NADH to generate excess superoxide radicals in the diabetic retina and cultured endothelial cells [24,25,26]. We believe that the discrepancies in results might be primarily due to the difference in the methodologies, as most of the investigators measured the metabolites and generation of free radicals either by using frozen tissues of diabetic animals or in the isolated diabetic retinas incubated without high glucose [8,22,27]. Since oxygen free radicals are extremely short-lived and their generation in an intact retina requires adequate oxygen tension in the incubation buffer. Due to these impediments, proper techniques and physiological conditions are warranted to measure the exact level of oxygen-free radicals in diabetic retinas.

Moreover, several researchers have reported excess free radicals being generated in cultured retinal cells under hyperglycemic conditions [7,9,28,29], which may not correlate with the results of the intact retina, since isolated cells cannot depict the exact pathophysiology as in the case of the whole retina; although increased ROS levels and their damaging effects are well known in the diabetic retina [9,30,31,32], their source and the mechanism of increasing ROS is still uncertain. Therefore, in this study, we adopted a unique experimental approach and techniques to measure precisely the generation of free radicals in a real-time situation under hyperglycemic and diabetic conditions in the intact rat retina. We analyzed the ROS generation in the control and diabetic retinas under in vivo and ex vivo experimental conditions, and a comparison was made between them to elucidate the basis of oxidative stress concerning hyperglycemia in diabetic retinopathy.

## 2. Materials and Methods

### 2.1. Animals

Wister albino rats were used in this study. Rats were housed under controlled conditions (25 °C; 12-h light–dark cycle) and allowed to have free access to food and water. The rats aged 8–9 weeks, weighing 260–290 g were used to make them diabetic using streptozotocin (STZ) from Sigma (St. Louis, MO, USA). A single intraperitoneal injection of STZ (65 mg/kg body weight) freshly prepared in 50 mM citrate-buffered solution (pH 4.5) was induced to each rat. Age-matched control rats were injected with an equal amount of the citrate buffer. Diabetes was confirmed by measuring blood glucose levels of more than 250 mg/dL. Retinal experiments were carried out either after 5 or 10 weeks of STZ-injections. Rats were routinely treated following the guidelines of the National Institutes of Health. All experimental procedures and protocols were under the Association for Research in Vision and Ophthalmology (ARVO) recommendations to the Care and Use of Experimental Animals. The experimental animal protocol has been approved by the Experimental Animal Care committee (approval number KSU-SE-21-04 ), King Saud University, Riyadh, Saudi Arabia.

### 2.2. Isolation of the Retinas and Incubation Conditions for Metabolic Studies

The 10 weeks STZ-diabetic and age-matched controls rats were anesthetized with ketamine-xylazine (53 mg ketamine, 5.3 mg xylazine/kg). Retinas from rats were dissected from excised eyes and the metabolic experimental protocol was followed according to our previously published methods with a slight modification [24]. [U-^14^C]glucose was purchased from New England Nuclear Life Science Products (Boston, MA, USA) for metabolic studies. First, each freshly excised intact retina from control and diabetic rats was preincubated for 3 min at 37 °C in 1 mL Krebs bicarbonate buffer (20 mM HEPES, pH 7.4, 118 mM NaCl, 4.7 mM KCl, 2.5 mM CaCl_2_, 1.2 mM KH_2_PO_4_, 1.17 mM MgSO_4_, 25 mM NaHCO_3_, with 5 or 20 mM glucose, equilibrated with 95% O_2_-5% CO_2_, pH 7.4) to metabolically recover the retina after removal from the animals [33]. Incubation was initiated by the addition of approximately 5 μCi [U-^14^C]glucose and terminated at 30 min by the addition of 20% perchloric acid (final concentration 2%). To evaluate CO_2_ and glutamate formation, incubation in the buffer was carried out under euglycemic (5 mM glucose) or hyperglycemic (20 mM glucose) conditions. A total of five control and diabetic rats were used in this study.

### 2.3. Oxidation of Glucose to CO_2_

To measure the oxidation of glucose to CO_2_, retinas from control and diabetic rats were incubated in 1 mL of Krebs buffer under euglycemic and hyperglycemic conditions at 37 °C in glass vials with the addition of [U-^14^C]glucose as described above, and a trap containing fluted filter paper was inserted in the vials. Immediately, the vials were sealed from the atmosphere. After 30 min, reactions were stopped by injecting 100 µL of 20% perchloric acid into the incubation buffer, and 100 µL of 1 N NaOH in the traps. ^14^CO_2_ formed by glucose oxidation reaction was allowed to diffuse out of the acidified samples and trapped in the filter paper traps soaked with NaOH. The filter paper traps were immersed in liquid scintillation fluid and counted after shaking for several hours. The disintegrations per minute of trapped ^14^CO_2_ in the filter paper are divided per milligram of retinal protein and by the specific activity of ^14^C-glucose to get values for ^14^CO_2_ formation per minute per milligram of protein.

### 2.4. Oxidation of Glucose as a Measure of Glutamate Formation

After incubating with [U-^14^C]glucose in the control and diabetic retinas, reactions were stopped by adding perchloric acid as described above. Retinas were homogenized and centrifuged to separate precipitated protein and the supernatant containing [^14^C]glutamate. The supernatant was neutralized and chromatographed using Dowex-1 acetate columns to separate glutamate with acetic acid [33]. The eluted [^14^C]glutamate from the column was quantitated by scintillation counting. The radiolabeled ^14^C-glutamate counts per minute divided by milligrams of retinal protein and by the glucose-specific activity permits to calculate the glutamate formation from glucose. Protein pellets obtained after centrifugation of retinal extract were sonicated in NaOH (0.5 mL of 1 M) and assayed for protein using the Bio-Rad reagent.

### 2.5. The Rate of H_2_O_2_ Clearance in the Excised Control and Diabetic Rat Retinas

The rate of intracellular H_2_O_2_ formation depends upon pro-oxidant superoxide dismutase (SOD), while its removal depends upon antioxidant catalase and peroxidases. H_2_O_2_ formed by SOD is removed by the catalase and peroxidases, which convert it into water. The rates of disposal of H_2_O_2_ by antioxidant enzymes were determined under ex vivo conditions in the excised whole retina of 5 and 10 weeks diabetic and age-matched control rats. Each of the freshly excised retinae from both groups of rats was first preincubated for 3 min at 37 °C in glass vials with 600 µL Krebs bicarbonate buffer, pH 7.4 containing 20 mM HEPES, 118 mM NaCl, 4.7 mM KCl, 2.5 mM CaCl_2_, 1.2 mM KH_2_PO_4_, 1.17 mM MgSO_4_, 25 mM NaHCO_3_, 5 mM glucose equilibrated with 95% O_2_-5% CO_2_ to allow the retina to adapt to the buffer. After 3 min, the preincubated buffer was replaced with 600 µL fresh Krebs bicarbonate buffer with either 5 mM glucose (euglycemic) in case of control retinas and 20 mM glucose (hyperglycemic) for diabetic retinas. The reaction was allowed until 30–40 min with the addition of 5µM H_2_O_2_. At every 5–10 min intervals, an aliquot of 50 µL was collected from each incubation vial to assay for H_2_O_2_ using Fluoro H_2_O_2_^TM^ kit (Cell Technology, Mountain View, CA, USA), following the company instructions. The H_2_O_2_ kit employs a non-fluorescent reagent to be oxidized by H_2_O_2_ to produce a fluorescent product, resorufin. The collected aliquot samples were assayed fluorometrically using excitation at 570, and emission at 590 nm wavelengths with a plate reader (Spectra-Max Plus; Molecular Devices, Sunnyvale, CA, USA).

In another set of experiments, an inhibitor of catalase, 3-aminotriazole (3-AT) was used to influence the rate of disposal of H_2_O_2_ by the excised control and diabetic retinas. Each retina was preincubated with 2 mM 3-AT in the Krebs bicarbonate buffer, 30 min before the addition of H_2_O_2_. The reaction was initiated with the addition of 5 µM H_2_O_2_ in the incubation buffer and at every 5–10 min intervals, 50 µL aliquots were collected from the buffer to measure the concentration of H_2_O_2_. After completion of the reactions, retinas were sonicated in 1 mL 50 mM phosphate buffer, pH 7.0 containing 0.1% SDS, and then centrifuged to obtain a supernatant. Total protein in the supernatant was measured using the Lowry method [34]. Disposal rates of H_2_O_2_ by the retina are expressed as % of H_2_O_2_ disposal/mg of protein.

### 2.6. The Measurement of the Level of H_2_O_2_ in the Excised Control and Diabetic Retinas

The level of H_2_O_2_ was determined under ex vivo conditions in the excised retinas from 10 weeks diabetic and control rats, under euglycemic and hyperglycemic conditions. Each of the freshly excised retinae from both groups of rats was separately preincubated at 37 °C in glass vials with 600 µL Krebs bicarbonate buffer as described above, containing 5 or 20 mM glucose equilibrated with 95% O_2_-5% CO_2_. In separate experiments, retinas were treated with 10 µM CuSO_4_. CuSO_4_ is known to catalyze the production of H_2_O_2_ and lowers the activity of catalase and glutathione peroxidase [35,36]. Aliquots of 100 µL from the reaction vials were collected after 15 and 30 min of incubation to measure the H_2_O_2_ generation in the retinas using the Fluoro H_2_O_2_ kit. After reactions, retinas were processed for protein estimation as described above. Results from H_2_O_2_ generation in the retina were presented as relative fluorescence unit/mg of protein.

### 2.7. The Measurement of ROS in the Excised Control and Diabetic Rat Retina

The fluorogenic marker CM-H_2_DCFDA (molecular probe) that passively diffuses into cells was used to measure ROS generation in the retina. Oxidation of CM-H_2_DCFDA yields fluorescent adducts that are trapped inside the cell. Fluorescent assay of the intracellular adducts provides a measure of ROS. Thus, the level of ROS was determined in the excised retinas from 10-week diabetic and age-matched control rats. Each of the freshly excised retina from diabetic and control rats was incubated at 37 °C in glass vials with 1 mL Krebs bicarbonate buffer equilibrated with 95% O_2_-5% CO_2_, along with 5 or 20 mM glucose and freshly made 10 µM CM-H_2_DCFDA. After 30 and 60 min of incubations, the retinas were separated and washed in cold 50 mM phosphate buffer saline. Then, those retinas were briefly sonicated in 300 µL 20 mM HEPES buffer, pH 7.4 containing 0.1% SDS. The retinal homogenate was centrifuged, and 100 µL supernatant immediately assayed fluorometrically at excitation and emission wavelengths of 485 and 538 nm, respectively. The level of ROS in the retina was presented as oxidized H_2_DCFDA fluorescence units/retina.

### 2.8. The Measurement of ROS under In Vivo Conditions in the Rat Retina

To measure the level of ROS in the 10 weeks diabetic and age-matched control retinas of live rats, a fresh stock solution (2.16 mM) of CM-H_2_DCFDA was made in DMSO, and 3 µL of the dye was injected intravitreally into the eye cavities of anesthetized rats according to our recently published method [37]. After six hours of injections, rats were anesthetized, retinas dissected, and immediately washed with cold phosphate buffer saline. Then, the retinas were homogenized by sonication in 300 µL of 20 mM HEPES buffer, pH 7.4 containing 0.1% SDS. The retinal homogenate was centrifuged, and 100 µL supernatant assayed fluorometrically. A comparison of the in vivo ROS level was made between the control and diabetic retina.

Additionally, we made three groups of control rats. In the first group, only 5 µL (2 µL saline + 3 µL of CM-H_2_DCFDA) was intravitreally injected into the retina. In the second group of rats, we intravitreally injected lipopolysaccharide (LPS, 1 µg/2 µL; plus, CM-H_2_DCFDA, 3 µL), and in the third group, diamide (2 µL/eye, 1 mM; plus, CM-H_2_DCFDA 3 µL) was injected. Lipopolysaccharide (LPS) is a well-known endotoxin to causes inflammation and increases the ROS level. Diamide is also known to increase oxidative stress by oxidizing glutathione [38]. After injections, the three groups of rats were housed overnight. After 16 h of injection, they were anesthetized, retinas dissected, washed in cold phosphate buffer saline, and sonicated in the 20 mM HEPES buffer, pH 7.4 containing 0.1% SDS, and processed as described above to assay the oxidized H_2_DCFDA fluorescence in each retina. Total retinal protein in the supernatant of each retina was measured. The level of oxidized fluorescence reflected the level of ROS in the retina, which is presented as fluorescence units/mg of retinal protein. The extent of the fluorescence level in the retina of three groups of control rats injected with; H_2_DCFDA alone, H_2_DCFDA + LPS, and H_2_DCFDA + diamide, were compared.

### 2.9. Statistical Analysis

Data are presented as means ± standard error of the mean (SEM). *p*-values less than 0.05 were considered significant. Statistical analyses were conducted by an unpaired, two-tailed Student *t*-test.

## 3. Results

### 3.1. Glucose Oxidation under Ex Vivo Condition in the Control and Diabetic Retina

We analyzed the influence of hyperglycemia and diabetes on flux through the citric acid cycle by measuring CO_2_ and glutamate production. The production of ^14^CO_2_ from [U-^14^C]glucose was measured in the 10 weeks control and diabetic rat retinas incubated with 5 or 20 mM glucose, respectively (Figure 1A). The rate of ^14^CO_2_ production was significantly decreased in diabetic rat retinas compared to controls when exposed to either 5 or 20 mM glucose (*p* < 0.01). Interestingly, there was also no significant influence of hyperglycemia on the rates of CO_2_ production in the controls compared to euglycemic exposure. Similarly, no significant change was observed between hyperglycemic and euglycemic diabetic retinas. Furthermore, the rate of [^14^C]glutamate formation modestly decreased in diabetic retinas compared to controls both under euglycemic and hyperglycemic conditions as shown in (Figure 1B); besides, no significant difference in the rate of glutamate formation in the control or diabetic retinas was observed under hyperglycemic and/or euglycemic conditions. The rate of glutamate formation reflected the differences seen in CO_2_ production. Both CO_2_ and glutamate data are related because both are tricarboxylic acid cycle fluxes. Therefore, despite the excess glucose in the diabetic retinas, they oxidized less glucose to CO_2_ and glutamate as compared to euglycemic controls.

### 3.2. Rates of Clearance of H_2_O_2_ under Ex Vivo Condition in the Control and Diabetic Retinas

We measured the rate of clearance of H_2_O_2_ in the 5 and 10-week diabetic and age-matched control rat retinas. First, we optimized the concentration of H_2_O_2_ for the clearance experiments, and 5 µM of H_2_O_2_ was found to be appropriate, as this concentration did not saturate under our experimental conditions. After applying 5 µM of H_2_O_2_ to 5 weeks excised control and diabetic retinas incubated under euglycemic and hyperglycemic conditions respectively, the level of H_2_O_2_ started to disappear linearly for at least 10 min in both groups. There was an insignificant difference in the H_2_O_2_ disposal between control and diabetic retina (Figure 2). The H_2_O_2_ clearance followed the first-order kinetics with an apparent 14 min half-life as calculated from the semi-logarithmic plot of the data (Figure 2, Insert). Furthermore, to analyze the influence of the duration of diabetes on the rates of H_2_O_2_ disposal, 10 weeks hyperglycemic-diabetic, and age-matched control rat retinas were employed. The rates of disappearance of H_2_O_2_ indicated a slight increase in the disposal rate of 10 weeks of diabetic retinas compared to euglycemic controls (Figure 3). The slope of the straight lines obtained from the logarithmic plot indicated rates of H_2_O_2_ clearance in the euglycemic control and hyperglycemic diabetic rat retinas (Figure 3, Insert). In the absence of retina but under similar conditions, the concentration of H_2_O_2_ in the incubation buffer remained constant for at least 40 min. An inhibitor of catalase, 3-aminotriazole (3-AT) was used to discriminate between the involvement of the two groups of antioxidant enzymes (catalase and glutathione peroxidase) for detoxification of H_2_O_2_ in both 5 and 10 weeks, control and diabetic retinas [39]. Surprisingly, no significant influence of catalase inhibitor on the rates of H_2_O_2_ disposal was observed in all the groups of control and diabetic rat retinas.

### 3.3. H_2_O_2_ Levels under Ex Vivo Conditions in Control and Diabetic Rat Retinas

We were not successful in the measurement of the level of H_2_O_2_ in the control and 10 weeks of diabetic rat retinas under ex vivo conditions even after 30 min of incubation under hyperglycemic conditions. However, a robust increase in the level of H_2_O_2_ was detected in the incubation buffer when retinas were treated with 10 µM CuSO_4_. A significant increase in the level of H_2_O_2_ was observed in the hyperglycemic diabetic retinas compared to euglycemic controls as soon as after 15 min of CuSO_4_ treatments (Figure 4). Moreover, the level of H_2_O_2_ did not increase further and remained persistent until 30 min of CuSO_4_ treatments, indicating a complete inactivation of antioxidant enzymes in the retina within 15 min of CuSO_4_ treatments.

### 3.4. ROS Levels under Ex Vivo Conditions in the Rat Retinas

The level of ROS was measured using CM-H_2_DCFDA dye (10 µM) in the excised control and 10 weeks diabetic retina, under euglycemic and hyperglycemic incubation conditions. We measured oxidized H_2_DCFDA fluorescence in the retina. As shown in Figure 5, there was a low endogenous ROS level detected in the hyperglycemic diabetic retinas after 30 and 60 min of incubation as compared to euglycemic controls. However, the difference in fluorescence was more evident after 60 min in the hyperglycemic diabetic retina compared to euglycemic controls. Contrary to several previous studies, our study shows that hyperglycemia seems to influence a decrease in the ROS level in the diabetic retina compared to euglycemic control retinas.

### 3.5. ROS Level under In Vivo Conditions in the Control and Diabetic Rat Retina

To measure the ROS production in retinas under in vivo conditions, we employed intravitreal injection of the “precursor” dye, carboxy-H_2_DCFDA. The dye passively diffused into the rat retina. The intracellular ROS formed, oxidized the trapped precursor dye in rat retinas which was measured as described in the above method section. The relative fluorescence unit was considered to be proportional to the level of ROS [40]. The fluorescence data from 10 weeks control and diabetic rat retinas are presented in Figure 6. The relative fluorescence was found to be more than 2-fold in the diabetic retinas as compared to controls. To validate this in vivo measurement of ROS in the rat retinas, we injected LPS and diamide as positive controls. Indeed, after 16 h of injection, both LPS and diamide caused a significant increase in the level of ROS as reflected by an increase in the oxidized dye fluorescence trapped inside the retina compared to only dye-injected retinas.

## 4. Discussion

The purpose of this study was to investigate oxidative stress in the rat retinas due to hyperglycemic and diabetic conditions that may cause long-term retinal damage, leading to diabetic retinopathy. To achieve this, we first studied the glucose oxidation to CO_2_ and glutamate under ex vivo conditions in the excised control and diabetic rat retinas using radiolabeled ^14^C-glucose. We measured the rate of CO_2_ and glutamate formation in retinas, which gives a measure of the rate of flux through the citric acid cycle. Secondly, we measured the antioxidant activity by hydrogen peroxide disposal and free radical generation in the excised intact retinas from control and diabetic rats under ex vivo experimental conditions, treated with euglycemic and hyperglycemic conditions. Finally, we employed in vivo techniques to analyze free radical generation in both control and diabetic rats by intravitreal injection of fluorogenic cell-permeant marker CM-H_2_DCFDA, as the oxidized fluorescent product of the dye gives a measure of intracellular level of ROS generation in the retina.

Several investigators proposed that a high serum level of glucose in diabetes increases intracellular levels of glucose, which in turn increases the glucose metabolism by inducing the rate of glycolysis. This is followed by an increase in citric acid fluxes that consecutively floods the mitochondria with excess reduced electron carriers (NADH) to increase the accumulation of ROS [7,28,29]. This mechanism of hyperglycemia-induced excess ROS generation has been widely accepted. However, our previous metabolic studies in the ex vivo rat retinas, using unique radio-isotopic techniques, indicated a decreased flux of glycolytic and citric acid cycle intermediates in diabetic retinas, which did not support an increase in ROS by mitochondria under hyperglycemic conditions [24]. We and others have been using the ex vivo retina or tissues, especially for metabolic studies, for a long time which is quite recognized in the field. Similarly, in this study, we measured glucose oxidation and oxidative stress parameters in the ex vivo retina of control and diabetic rats. We found a decreased rate of glucose oxidation, as evidenced by a reduced level of CO_2_ and glutamate in the diabetic retina. This indicates that the mitochondrial electron transport chain may not be under the influence of high electron pressure to escape electrons to make excess ROS. Thus, our studies negate the generation of excess ROS through mitochondria under hyperglycemic conditions in the diabetic rat retinas as opposed to several previous studies [8,9,11,16,17,18,19]. Generally, mitochondria through the electron transport system generate a major part of cellular ROS, but the production is low under normal conditions. However, due to the excess level of NADH, some electrons released might not get reduced to O_2_ and H_2_O. Thus, these escaped electrons generate superoxide and oxygen free radicals [20,21]. Moreover, if oxygen free radicals are generated in excess, they are instantly detoxified by mitochondrial antioxidant enzymes to harmless products. Hydrogen peroxide, a powerful oxidizing agent which is generated by mitochondrial superoxide dismutase can be detoxified by catalase and peroxidase enzymes to water molecules. In this study, we measured the antioxidant capacity of these enzymes in terms of H_2_O_2_ disposal in the retina of two aged groups (5 and 10 weeks) of control and STZ-diabetic rats under ex vivo conditions. Interestingly, we found a similar activity of H_2_O_2_ detoxifying enzymes in the control and diabetic retinas from both groups of rats even after diabetes is prolonged. Thus, contrary to a few previous studies [8,16,17,41,42,43], our study suggests that there is little to no influence of diabetes or the short-term duration of diabetes (5–10 weeks) on the antioxidant capacity of mitochondrial enzymes, as evident from the rate of disposal of H_2_O_2_ in the rat retinas. This is partly supported by the Obrosova group, who reported that catalase activity was high, but not low in diabetic rat retina [44]. Moreover, after treating retinas with 3-aminotriazole (a specific inhibitor of catalase), no difference in the rates of H_2_O_2_ disposal was found between control and diabetic retinas of the two groups of rats. This suggests the possibility of a major role of peroxidases, other than catalase in the degradation of H_2_O_2_. A study by Makino et al. reported that glutathione peroxidase detoxifies H_2_O_2_ at a concentration below 10 µM, as we only used 5 µM H_2_O_2_ in our experiment, whereas catalase contributes at a higher concentration [45]. Our results further suggest the existence of a strong antioxidant system in the retina to detoxify excess H_2_O_2,_ if generated in the case of diabetic retinas. Also, the measurement of H_2_O_2_ level in the ex-vivo rat retinas indicated that the concentration of H_2_O_2_ was too little to be detected by our H_2_O_2_ kit. For this reason, we exposed the excised retina with CuSO_4_, a known inhibitor of catalase and peroxidases to induce the production of H_2_O_2_ [35,36]. Indeed, after exposing the retinas with CuSO_4_, a robust increase in H_2_O_2_ generation was observed. Interestingly, a significant increase in the level of H_2_O_2_ was observed in the hyperglycemic diabetic retinas compared to euglycemic controls, and the difference remained unchanged after prolonged incubation. These results suggest that antioxidant enzymes (catalase, peroxidases) became inactivated by CuSO_4_, but on the other side, SOD appears to be relatively activated in the diabetic retinas to generate an increased level of H_2_O_2_ as compared to non-diabetic controls. We speculate that the increase in H_2_O_2_ is not due to hyperglycemia-induced excess pressure on mitochondria, rather due to diabetes-induced non-mitochondrial sources such as by activation of xanthine oxidase, NADPH oxidase, and peroxisomes in the cell.

Next, we employed the fluorescent CM-H_2_DCFDA dye to analyze the ROS generation in the excised control and diabetic retina, under euglycemic and hyperglycemic incubation conditions. CM-H_2_DCFDA dye passively diffused inside the cells and the extent of the oxidized fluorescent product of the dye corresponding to the intracellular level of ROS generation [40]. Surprisingly, a significantly low endogenous ROS level was detected in the diabetic retinas under hyperglycemic conditions as compared to euglycemic controls. This is further supported by our recent in vitro studies using cultured rat retinal cells (Muller and endothelial cells), where we found a significantly low ROS level when cells were treated with high glucose (25 mM) as compared to euglycemic conditions (Unpublished data). This observation is supported by a few other studies reporting that pyruvate, the glycolytic product of glucose is a strong antioxidant and protects the retina and retinal cells under diabetic conditions [46,47]. Thus, contrary to several previous studies, hyperglycemia seems to influence a decrease in the ROS generation in the excised diabetic retina compared to euglycemic controls.

Our next aim was to measure the ROS generation under in vivo conditions in the intact control and diabetic rat retina by intravitreal injection of CM-H_2_DCFDA, for which we successfully adopted the method recently published [37]. As expected, when rats were intravitreally injected with LPS and diamide, which served as positive controls in this study, induced a significant increase in the ROS generation. In agreement with most of the studies, the in vivo ROS level in diabetic rat retinas was significantly high compared to controls. This increased generation of ROS in the diabetic retina indicates a possibility of either diabetes-induced activation of paracrine mediators or activation of non-mitochondrial oxidases that may influence the excess ROS generation [26,27].

Taken together, our data show that oxidation of glucose decreased in the diabetic retina despite hyperglycemic conditions. These decreased levels of oxidation of glucose in the diabetic retina indicate a slow rate of glycolysis and/or citric acid cycle, thereby suggesting that excess ROS may not be generated by mitochondria. The duration of diabetes and treatments to high glucose could not influence the retinal antioxidant capacity of mitochondrial enzymes in the disposal of H_2_O_2_, suggesting that mitochondria may not be a major source of oxidative stress in the diabetic retina. Nevertheless, an increased level of ROS was found under in vivo conditions in the diabetic retinas that indicate the possibility of non-mitochondrial sources of ROS generation, which may include activation of NADPH and NADH oxidases [27,48,49], activation of endothelial cells by paracrine mediators [25], activation of microglia [50], and glutamate excitotoxicity [51,52]. Thus, metabolic abnormalities by hyperglycemia per se, especially through mitochondrial stress may not be the sole basis of retinal damage in diabetic retinopathy. Besides diabetes-induced hyperglycemia, emerging evidence suggests a potential role of numerous other altered metabolites and factors that need to be considered in the pathophysiology of retinal damage through oxidative stress. In addition, further metabolic studies and possibly in vivo ROS imaging techniques are required to better elucidate the mechanism of ROS production and their major sources in the diabetic retina.

## Figures and Tables

**Figure 1 cells-10-00794-f001:**
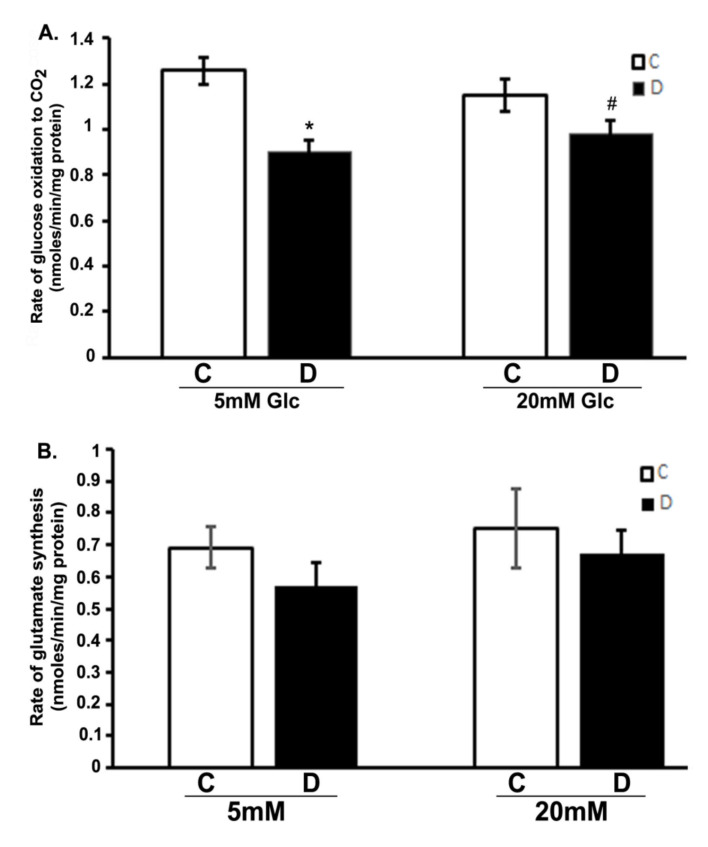
Glucose oxidation in the diabetic rat retinas. Retinas from 10-week diabetic rats and their controls were incubated in the buffer with 5 or 20 mM glucose at 37 °C for 30 min. Rates of the production of ^14^CO_2_ and ^14^C-glutamate in retinas were measured as described in the method section. (**A**). Oxidation of glucose to CO_2_ in diabetic retinas significantly decreased compared to control retinas (* *p* < 0.05, control vs. diabetic at 5mM glucose; and ^#^
*p* < 0.05, control vs. diabetic at 20 mM glucose). (**B**). Rate of glutamate synthesis in diabetic retina moderately decreased compared to controls incubated with 5 or 20 mM glucose. Rates of CO_2_ and glutamate were measured as nmoles/min/mg of retinal protein. Data are expressed as means ± SEM (n = 6 for each).

**Figure 2 cells-10-00794-f002:**
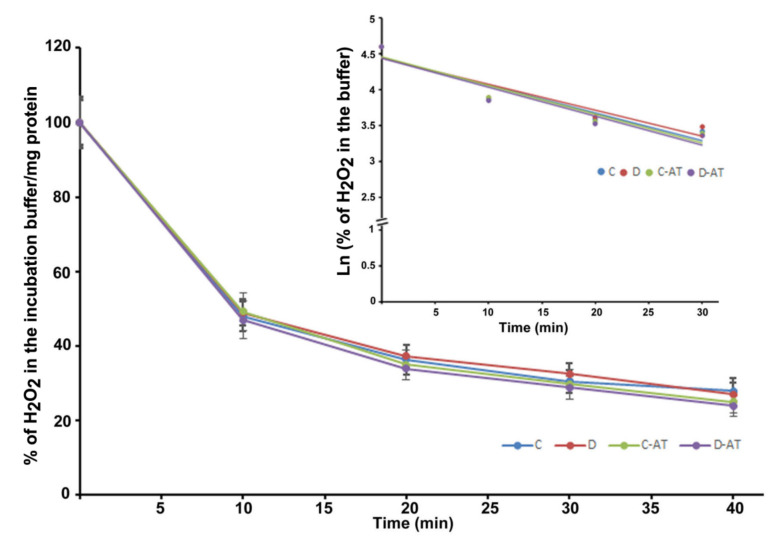
Clearance of H_2_O_2_ by the 5 weeks excised control and diabetic retina under euglycemic and hyperglycemic incubation conditions. Control retinas were incubated in the buffer with 5 mM glucose ± 3-aminotriazole (3-AT), and diabetic retinas with the buffer containing 20 mM glucose ± 3-AT. The retinas were pre-incubated for 30 min in the incubation buffer with or without 3-AT. Clearance reactions by the retinas started with the addition of 5 µM H_2_O_2_. Insert: semi-logarithmic representation of the data obtained during the 30 min period. The half-lives are approximately 14 and 15 min, in the control and diabetic retina, respectively. No significant change in the rate of disposal was observed in the case of 3-AT treatments in both control and diabetic retinas. The content of the retinal protein was determined for each incubation. Values are means ± SEM for 5 determinations (n = 5 for each). (C) control; (D) diabetic; AT (aminotriazole).

**Figure 3 cells-10-00794-f003:**
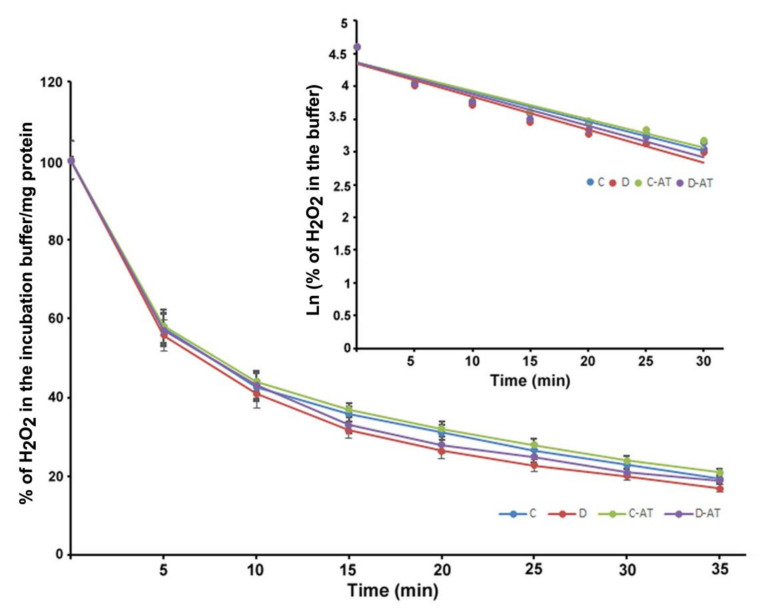
Clearance of H_2_O_2_ by 10 weeks excised control and diabetic retina under euglycemic and hyperglycemic incubation conditions. Control retinas were incubated in the buffer containing 5 mM glucose ± 3-AT and diabetic retinas with the buffer containing 20 mM glucose ± 3-AT. The retinas were pre-incubated for 30 min in the incubation buffer with or without 3-AT. Clearance reactions by the retinas started with the addition of 5 µM H_2_O_2_. Insert: semi-logarithmic representation of the data obtained during the 30 min period. The half-lives are approximately 10 and 9 min, in the control and diabetic retina, respectively. No significant change in the H_2_O_2_ disposal rate was observed in the case of 3-AT treatments in both control and diabetic retinas. Values are means ± SEM for 5 determinations (n = 5 for each).

**Figure 4 cells-10-00794-f004:**
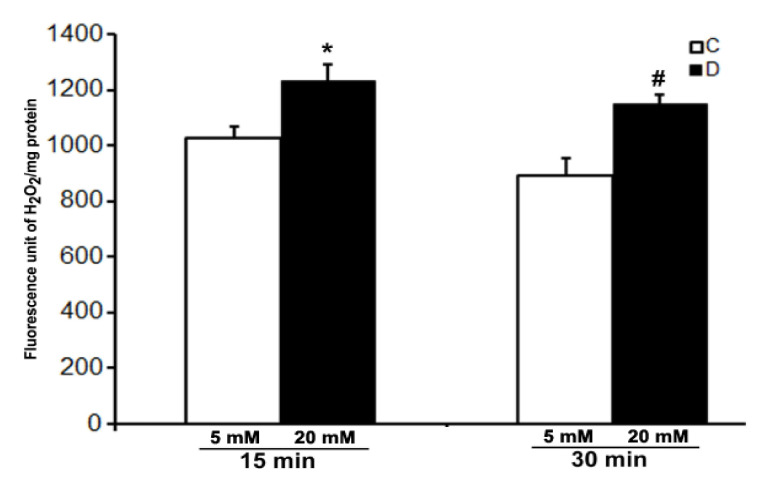
Effect of CuSO_4_ on the H_2_O_2_ level in excised control and diabetic rat retinas. 10-week diabetic retinas were incubated in Krebs bicarbonate buffer with 20 mM glucose, and their age-matched euglycemic controls were incubated with 5 mM glucose. H_2_O_2_ measurement was done in the buffer after 15 and 30 min of 10 µM CuSO_4_ treatments. The content of the retinal protein was determined for each incubation. Values are means ± SEM for 5 determinations (n = 5 for each). *^,#^
*p* < 0.05, diabetic versus control retina.

**Figure 5 cells-10-00794-f005:**
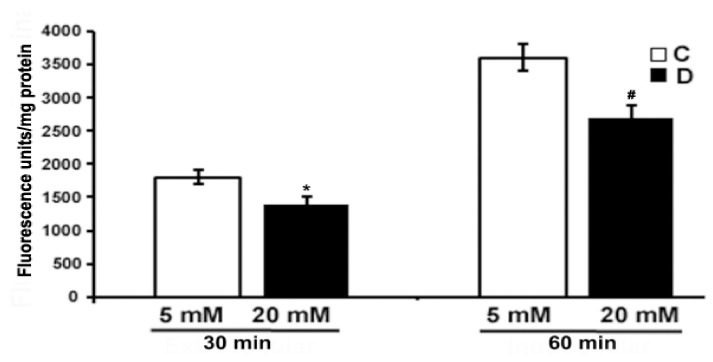
Measurement of oxidative stress in excised control and diabetic retina using fluorescent ROS indicator. Excised retinas from 10 weeks control and diabetic retinas were incubated in the Krebs bicarbonate buffer with 10 µM CM-H_2_DCFDA for 30 and 60 min. The fluorescence intensity of oxidized H_2_DCFDA was measured within the retina as described in the method section. Fluorescence from oxidized H_2_DCFDA in the retina is proportional to ROS. Data are expressed as means ± SEM (n = 5–6 for each). *^,#^
*p* < 0.05, diabetic versus control retina.

**Figure 6 cells-10-00794-f006:**
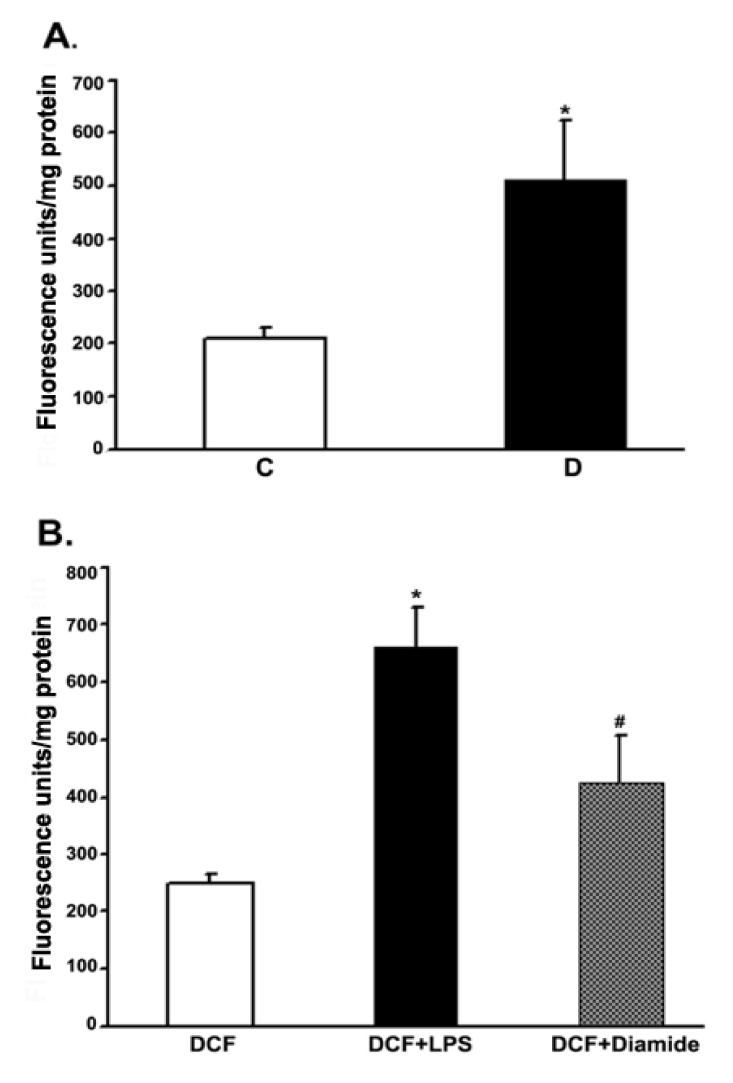
Measurement of the ROS levels in rat retinas under in vivo conditions. Rats were intravitreally injected with either CM-H_2_DCFDA or LPS or diamide as described in the method section. The extent of the oxidized fluorescence and retinal protein were measured in each excised retina, which is presented as fluorescence units/mg of retinal protein. (**A**). Control (C) vs. diabetic retina (D): * *p* < 0.01. (**B**). Control retina: * *p* < 0.01, H_2_DCFDA vs. (H_2_DCFDA + LPS); ^#^
*p* < 0.01, H_2_DCFDA vs. (H_2_DCFDA + Diamide). Values are expressed as means ± SEM (n = 5–6 for each).

## Data Availability

All relevant data are included within the manuscript. The raw data supporting the findings of this manuscript will be provided by the author to any researcher on reasonable request.

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
