# Peer review of "Does Hyperglycemia Cause Oxidative Stress in the Diabetic Rat Retina?"

_cells, 2021, doi:10.3390/cells10040794_

Round 1
Reviewer 1 Report
Herein, Dr. Ola explores the impact of STZ-induced diabetes and hyperglycemic culture conditions on oxidative stress in rat retina. The data are interesting and explore an important problem in the field, ie diabetes-induced oxidative stress. The data support that ex vivo retina from diabetic rats exhibits a reduced capacity to oxidize glucose and dispose of free radicals. This is in contrast to CuSO4-induced ROS levels, where the retina from diabetic rats exhibits increased free radical production, as assessed by DCF fluorescence. Surprisingly, there is no effect of low versus a high glucose concentration in the Krebs buffer used for retinal cultures on these endpoints. Moreover, the intact retina from STZ-diabetic rats did exhibit increased DCF fluorescence as compared to non-diabetic controls. Dr. Ola concludes that this may suggest a role for hyperglycemia-independent non-mitochondrial ROS as opposed to mitochondrial. Support for this should be more clearly articulated, as such a conclusion relies heavily on prior findings and is difficult to see based on the data herein alone. There are also a number of other significant issues that must be addressed:
- There is no significant difference in glucose oxidation to glutamate with diabetes (Fig 1B). To say that it “moderately decreased” does not accurately reflect the data (see line 229 and the figure legend). Perhaps an alternative statistical analysis could be made by collapsing the C and D retinas regardless of glucose concentration in the Krebs buffer.
- Appropriate statistical analysis of repeated measures does not appear to have been performed in Figs 2-3. It is neither indicated in the methods nor legends. Suggest editing the Y-axis label in Fig 2. I don’t think this is disposal per se, but rather the amount of hydrogen peroxide remaining relative to the start of the study. That may be an incorrect interpretation, and if so, I suggest editing the text to clarify. Also the Y-axis label in Fig 6 could also be improved for accuracy. I also don’t believe that Fig 5/6 measures “ROS generation” per se, but rather ROS levels. I’m not sure there’s any way to account for a difference in ROS disposal with DCF alone. Certainly variation in antioxidant levels have been shown to influence this endpoint.
- Figure 3: “Disposal of H2O2 in the 10 weeks excised control and diabetic retina…” To be clear, I believe that this in not IN the retina, but rather a measurement of the oxidant level IN the Krebs buffer. I think it would still be accurate to call this “disposal by the ex vivo retina”, but to imply that it is occurring intracellularly seems misleading. This important difference is likely responsible for the 3-AT results, as inhibition of the intracellular enzyme catalase does not appear to influence disposal rate.
- The ex vivo retina culture model is a major limitation of many of the key conclusions in the discussion. Glucose applied in the media may not be a good model for glucose in perfused retina vessels, particularly with regards to how higher glucose concentrations get into the cells. Such a limitation should be acknowledged in the discussion. What retinal cell types express GLUT2? Where are these high capacity transporters located?
- Are the data here consistent with SLGT-inhibition to lower glucose levels in the STZ rat? This would be a nice addition to Fig 6B.
Minor Comments-
Suggest editing the following- lines 23, 56, 82, 194/207, 236, 375
“H2O2” is often used without appropriate subscripts
Line 172: Is this kit proprietary? Manufacturer? How does it specifically detect hydrogen peroxide?
Line 297: *P<0.05, control vs diabetic at 5mM glucose; and *P<0.05 control vs. diabetic at 20 mM …. This is redundant, ie. *P<0.05, control versus diabetic. It is also unclear why two different symbols are used to indicate the same control versus diabetic comparison in Fig 4.
“3-aminotrizole”, misspelled?
A “robust increase” in hydrogen peroxide with copper sulfate exposure is not demonstrated (line 262). This data should either be included, or only differences in copper sulfate-induced should be noted.
The presentation of data in Fig 6A-B is presented in reverse order in the results section.
Reviewer 2 Report
In this work, the author investigated the sources of ROS in the retina and the role of oxidative stress in the development of diabetic retinopathy. There are a number of comments on it:
Major Concerns
- There is no normalization in a number of figures. It is not clear why the author does not recalculate the results for the absolute amount of H2O2 formed, but gives the units of H2O2 fluorescence per mg of protein. This also applies to other figures.
- It is not clear what is the meaning of the experiments in Fig. 4 and 5. The author used different times, different conditions, why was it done? This is not clear from the text. In fig. 5 the author needs to show hyperglycemic control and euglycemic diabetes. Why, in one case, the author used 15 minutes, and in the other 60 min, etc. The author needs to standardize the experimental conditions.
- The author indicates in the text that mitochondria are not the main producers of ROS under these conditions. However, this does not follow from the experimental data. I think that the author should think carefully about the design of the study in order to conclusively show the absence of a leading role of mitochondria in the development of oxidative stress in diabetic retinopathy.
Comments:
- The author claims that retinopathy is the leading cause of blinding disease worldwide. It is necessary to show the appropriate reference.
- 1B, according to the figure caption, the data differ, however, according to the figure, this is not the case, it is necessary to coordinate the figure and the caption.
- In this case, the authors also need to explain this discrepancy in the text. Why does the rate of CO2 formation decrease, but the formation of glutamate does not change?
Round 2
Reviewer 1 Report
All of my concerns have been addressed.
Reviewer 2 Report
I am generally satisfied with the changes made and have no more questions